# New Steroidal Saponins Isolated from the Rhizomes of *Paris mairei*

**DOI:** 10.3390/molecules26216366

**Published:** 2021-10-21

**Authors:** Yang Liu, Pengcheng Qiu, Minchang Wang, Yunyang Lu, Hao He, Haifeng Tang, Bang-Le Zhang

**Affiliations:** 1Department of Pharmaceutics, School of Pharmacy, Air Force Medical University, Xi’an 710032, China; so870823@163.com; 2Department of Chinese Materia Medica and Natural Medicines, School of Pharmacy, Air Force Medical University, Xi’an 710032, China; qpc023@126.com (P.Q.); luyunyanggq@163.com (Y.L.); 3Xi’an Modern Chemistry Research Institute, Xi’an 710065, China; wmc204@163.com; 4State Key Laboratory of Fluorine & Nitrogen Chemicals, Xi’an 710065, China; 5School of Pharmacy, Xi’an Medical University, Xi’an 710021, China; hehao313@163.com

**Keywords:** *Paris mairei*, steroidal saponins, spirostane saponin, 15-oxo-18-nor-spirost, cytotoxicity

## Abstract

The genus *Paris* is an excellent source of steroidal saponins that exhibit various bioactivities. *Paris mairei* is a unique species and has been widely used as folk medicine in Southwest China for a long time. With the help of chemical methods and modern spectra analysis, five new steroidal saponins, pamaiosides A–E (**1**–**5**), along with five known steroidal saponins **6**–**10**, were isolated from the rhizomes of *Paris mairei*. The cytotoxicity of all the new saponins was evaluated against human pancreatic adenocarcinoma PANC-1 and BxPC3 cell lines.

## 1. Introduction

The genus *Paris* (Liliaceae) includes 33 species around the world, and 27 species and more than 15 varieties have been discovered in China [1]. It has been used as a traditional Chinese medicine for traumatic injuries, heat-clearing and detoxifying, and relief of swelling and long-term pain [2]. Under the development of phytochemistry, steroidal saponins have been proved to be the main chemicals in the genus *Paris* and present a wide range of pharmacological activities such as anti-tumor [3,4,5], anti-inflammatory [6], anti-fungal [7], hemostasis [8], and immunomodulatory [9]. Moreover, *Rhizoma Paridis*, documented as rhizomes of *Paris polyphylla* var. *yannanensis* and *Paris polyphylla* var. *Chinensis* in the 2020 edition of the Chinese Pharmacopoeia, is usually used as adjuvant drugs for postoperative treatment of cancer to improve symptoms and therapeutic effect. However, *Paris polyphylla* var. *yannanensis* and *Paris polyphylla* var. *Chinensis* as perennial plants need at least 5 years to mature, and the increasing market demand makes wild sources of *Rhizoma Paridis* seriously scarce [10]. Hence, it is necessary to investigate other species of *Paris* in order to relieve resource pressure. *Paris mairei* is mainly distributed in the Guizhou, Sichuan, and Yunnan provinces of China and used as folk medicine for a long time. Herein, this paper reports the isolation and structural identification of five new (**1**–**5**) and five known (**6**–**10**) saponins (Figure 1) as well as the cytotoxicity against human pancreatic adenocarcinoma PANC-1 and BxPC3 cell lines.

## 2. Results and Discussion

Compound **1**, named Pamaiosides A, a white amorphous solid, was positive to Liebermann Burchard and Molisch chemical reactions, which indicates that it might be a steroidal glycoside. The pseudomolecular ion peak was detected in the HR-ESI-MS spectrum at *m/z* 995.4824 [M + Na]^+^ (calculated for C_48_H_76_O_20_Na, 995.4828), corresponding to the molecular formula C_48_H_76_O_20_. Four methyl groups were tested in the ^1^H-NMR spectrum at *δ*_H_ 0.82 (3H, *s*, H-18), 1.12 (3H, *s*, H-19), *δ*_H_ 0.80 (3H, *d*, *J* = 6.35 Hz, H-27), and 0.96 (3H, *d*, *J* = 6.90 Hz, H-21). Meanwhile, one olefinic methine proton signal was observed at *δ*_H_ 5.56 (1H, *br s*, H-6). The hydrogen signals above suggest a steroid skeleton [11,12]. Correspondingly, in the ^13^C-NMR spectrum of **1**, four carbon signals of methyl groups were revealed at *δ*_C_ 17.65 (C-18), 15.52 (C-19), 17.29 (C-27), and 15.07 (C-21) as well as one trisubstituted double bonds at *δ*_C_ 139.71 (C-5) and 126.17 (C-6). A characteristic hemiacetal signal of spirostanol aglycone was discovered at *δ*_C_ 111.74 (C-22) [11]. In the HMBC spectrum, the cross-peaks between H-4 (*δ*_H_ 1.90) and C-5 (*δ*_C_ 139.71), H-19 (*δ*_H_ 1.12) and C-5 (*δ*_C_ 139.71), and H-6(*δ*_H_ 5.56) and C-8 (*δ*_C_ 34.26)/C-10 (*δ*_C_ 43.58) inferred that the double bond was located at C-5/C-6 (Figure 2). In the NOESY spectrum, the correlation between H-1 (*δ*_H_ 3.37) and H-9 (*δ*_H_ 1.25) and H-3 (*δ*_H_ 3.34) and H-9 (*δ*_H_ 1.25) suggested that the configurations of H-1 and H-3 were an *α*-orientation, so that the hydroxyl substituent at C-1 and C-3 were both *β* configuration. The correlation between H-3 (*δ*_H_ 3.34) and H-16 (*δ*_H_ 4.38)/H-17 (*δ*_H_ 1.72), between H-16 (*δ*_H_ 4.38) and H-17 (*δ*_H_ 1.72), between H-8 (*δ*_H_ 1.56) and H-18 (*δ*_H_ 0.82), between H-19 (*δ*_H_ 1.12) and H-11 (*δ*_H_ 1.42), and between H-9 (*δ*_H_ 1.25) and H-14 (*δ*_H_ 1.15) elucidate the usual *trans* junction for the B/C and C/D rings. The correlations between H-8 (*δ*_H_ 1.56) and H-20 (*δ*_H_ 1.90) infer that C-20 was an *S* configuration. In the spirostanol saponins, when the resonance of the proton H-20 was observed at a lower field than approximately δ_H_ 2.48, the orientation relationship between the proton of H-20 and the oxygen atom included in the F ring was thought to be located at the *cis* position. On the other hand, when the proton shifts of H-20 were detected at a higher field than δ_H_ 2.20, the orientation relationship is thought to be *trans* [13,14]. In this way, the orientation relationship of the F ring was considered to be *trans,* and the configuration of C-22 was confirmed as *R.* The 25*R* configuration was determined by the chemical shift difference between H-26a and H-26b (∆ = *δ*_Ha_ *−*
*δ*_Hb_ = 3.43 − 3.30 = 0.13 < 0.48) [15,16]. By combining the data and consulting the literature [17], the aglycone of compound **1** was identified as (20*S*,22*R*,25*R*)-spirost-5-en-1*β,*3*β*-diol.

According to the ^13^C-NMR spectrum, except for the 27 signals of aglycone, the remaining 21 belonged to the oligosaccharide’s moiety. After acid hydrolysis and derivatization with N-(trimethylsilyl) imidazole, the derivates were compared with retention times to the corresponding authentic samples by GC analysis; thus, the monosaccharide residues were identified as L-Ara, L-Rha, D-Xyl, and D-Api in a ratio of 1:1:1:1. In the ^1^H-NMR spectrum, four anomeric proton signals were obvious at *δ*_H_ 4.34 (*d*, *J* = 7.35 Hz, H-1 of Ara), *δ*_H_ 5.31 (*b*r *s*, H-1 of Rha), *δ*_H_ 4.41 (*b*r *d*, *J* = 7.1 Hz, H-1 of Xyl), and *δ*_H_ 5.19 (*d*, *J* = 2.9 Hz, H-1 of Api). The corresponding carbon signals were successfully searched at *δ*_C_ 101.16, *δ*_C_ 101.60, *δ*_C_ 106.47, and *δ*_C_ 112.17 in the HSQC spectrum, respectively. By analyzing the ^1^H-NMR, TOCSY, and HSQC spectra, the sequence and location of protons and carbons were determined in each monosaccharide (Table 1, Table 2, Table 3 and Table 4). The sequence of a tetrasaccharide chain was confirmed by the HMBC spectrum, which acted as the correlations from Rha H-1 (*δ*_H_ 5.31) to Ara C-3 (*δ*_C_ 80.45), Api H-1 (*δ*_H_ 5.19) to Xyl C-4 (*δ*_C_ 70.54), Xyl H-1 (*δ*_H_ 4.41) to Ara C-4 (*δ*_C_ 85.29), and the cross-peak between Ara H-1 (*δ*_H_ 4.34) and C-1 (*δ*_C_ 84.79) demonstrated the location of a sugar linkage. The anomeric proton coupling constants of D-xylopyranose (*J* = 7.1 Hz > 7.0 Hz) and L-arabopyranose (*J* = 7.35 Hz > 7.0 Hz) suggested that the configurations had a *β*-orientation and an *α*-orientation, respectively [18,19]. The *β* configuration of D-apiose was determined by the chemical shifts of *δ*_C_ 112.17 (C-1), *δ*_C_ 78.23(C-2)*,*
*δ*_C_ 80.49(C-3), *δ*_C_ 75.18 (C-4), and *δ*_C_ 65.56 (C-5) [20]; the *α* anomeric configuration of L-rhamnopyranosyl was confirmed by the chemical shifts of Rha C-5 at *δ*_C_ 69.84 [21]. Thus, the structure of Pamaiosides A (**1**) was characterized as (20*S*,22*R*,25*R*)-spirost-5-en-1*β*,3*β*-diol-1-*O*-*β*-d-apiofuranosyl-(1→4)-*β*-d-xylopyranosyl-(1→4)-[*α*-l-rhamnopyranosyl-(1→3)]-*a*-l-arabinopyranoside.

Compound **2**, named Pamaiosides B, a white amorphous solid, was positive to Liebermann Burchard and Molisch chemical reactions. The pseudomolecular ion peak was measured in the HR-ESI-MS spectrum at *m/z* 1043.4677 [M + Na]^+^ (calculated for C_48_H_76_O_23_Na, 1043.4675), corresponding to the molecular formula C_48_H_76_O_23_. Compared to **1**, one angular methyl at *δ*_C_ 15.07 (C-21) and two methylenes at *δ*_C_ 32.59 (C-23) and 30.04 (C-24) were absent, and the chemical shifts were all markedly up-field at *δ*_C_ 62.95 (C-21, ∆*δ*_C_ + 47.88 ppm), *δ*_C_ 71.20 (C-23, ∆*δ*_C_ + 38.61 ppm), and *δ*_C_ 74.01 (C-24, ∆*δ*_C_ + 43.97 ppm), respectively, which indicates that a hydroxyl group substituted at the primary carbon atom (Table 1, Table 2, Table 3 and Table 4). In the HMBC spectra, the cross-peaks between *δ*_H_ 2.78 (H-20) and *δ*_C_ 62.95 (C-21) and between *δ*_Ha_ 3.55, *δ*_Hb_ 3.69 (H-21) and *δ*_C_ 46.04 (C-20)/*δ*_C_ 112.72 (C-22) authenticated the hydroxyl substituted at C-21, and it was further confirmed by the correlations for *δ*_Ha_ 3.55, *δ*_Hb_ 3.69 (H-21) to *δ*_H_ 2.78 (H-20) in the ^1^H-^1^H COSY spectrum (Figure 3). Meanwhile, the correlations for *δ*_H_ 1.91 (H-25) to *δ*_H_ 3.76 (H-24) and *δ*_H_ 3.76 (H-24) to *δ*_H_ 3.52 (H-23) in the ^1^H-^1^H COSY spectrum and the signal of *δ*_H_ 0.90 (H-27) to *δ*_C_ 74.01 (C-24) derived from the HMBC spectrum illustrated that the hydroxyl displaced at C-23 and C-24. In the NOESY spectra, the configurations of C-1, C-3, C-23, and C-24 were successively evidenced as *β*, *β*, *α*, and *β* orientations derived from correlations for H-1 (*δ*_H_ 3.40) to H-9 (*δ*_H_ 1.25), H-3 (*δ*_H_ 3.38) to H-9 (*δ*_H_ 1.25), H-20 (*δ*_H_ 2.78) to H-23 (*δ*_H_ 3.52), and H-24 (*δ*_H_ 3.76) to H-27 (*δ*_H_ 0.90), respectively. Using the same method as for **1**, C-20, C-22, and C-25 were determined as *R* configuration. By summarizing the data and comparing it to the literature [22], the aglycone of compound **2** was established as (20*R*,22*R*,25*R*)-spirost-5-en-1*β*,3*β*,21,23*α*,24*β*-pentol.

Acid hydrolysis, derivatization, and GC analysis revealed that compound **2** possessed the same monosaccharide residues as **1**, but different linkages emerged between the sugars. In the HMBC spectra, the sugar sequencing linkages were testified by the correlations between Api H-1 (*δ*_H_ 5.21) and Rha C-3 (*δ*_C_ 80.45), Rha H-1 (*δ*_H_ 5.33) and Xyl C-2 (*δ*_C_ 74.89), Xyl H-1 (*δ*_H_ 4.43) and Ara C-3 (*δ*_C_ 85.25), and Ara H-1 (*δ*_H_ 4.34) and C-1 (*δ*_C_ 84.80). Thus, compound **2** was elucidated as (20*R*,22*R*,25*R*)-spirost-5-en-1*β*,3*β*,21,23*α*,24*β*-pentol-1-*O*-*β*-d-apiofuranosyl-(1→3)-*α*-L-rhamnopyranosyl-(1→2)*-β*-d-xylopyranosyl-(1→3)-*a*-L-arabinopyranoside.

Compound **3**, named Pamaiosides C, a white amorphous solid, was positive to Liebermann Burchard and Molisch chemical reactions. The pseudomolecular ion peak was measured in the HR-ESI-MS spectrum at *m/z* 993.3932 [M + Na]^+^ (calculated for C_46_H_66_O_22_Na, 993.3943), corresponding to the molecular formula C_46_H_66_O_22_. Compared to **2**, one angular methyl *δ*_C_ 17.30 (C-18) was missing and two quaternary carbons, *δ*_C_ 179.23 (C-13) and 139.51 (C-14), and one ketone *δ*_C_ 207.09 (C-15) signal were detected (Table 1, Table 2, Table 3 and Table 4). In the HMBC spectra, the cross-peaks between *δ*_Ha_ 1.19, *δ*_Hb_ 2.94 (H-11)/*δ*_Ha_ 2.36, *δ*_Hb_ 2.60 (H-12)/*δ*_H_ 2.34 (H-17)/*δ*_H_ 4.38 (H-16) and *δ*_C_ 179.23 (C-13), between *δ*_H_ 2.26 (H-8)/*δ*_Ha_ 1.48, *δ*_Hb_ 2.87 (H-7), and *δ*_C_ 139.51 (C-14) allowed to deduce that one double bond was located at C-13/C-14. Moreover, the location of *δ*_C_ 207.09 (C-15) was affirmed by correlation of *δ*_H_ 2.34 (H-17) to *δ*_C_ 207.09 (C-15) (Figure 4). As a result, the aglycone of **3** was determined as 15-oxo-18-nor-(20*R*,22*R*,25*R*)-spirost-5,13-diene-1*β*,3*β*,21,23*α*,24*β*-pentol [23].

The monosaccharide residues were identified as L-Ara, L-Rha, and D-Api in a ratio of 1:1:1 by acid hydrolysis, derivatization, and GC analysis. In addition, two keto-methyls at *δ*_C_ 21.15 (3H, *s*, *δ*_H_ 2.12) and 21.02 (3H, *s*, *δ*_H_ 2.02) and two carbonyl at *δ*_C_ 172.29 and 171.96 carbons signals were observed, which infers that two acetyl groups existed in the sugar chain. In the HMBC spectrum, one proton of keto-methyl at *δ*_H_ 2.12 was correlated with one carbonyl carbon signal at *δ*_C_ 172.29 and *δ*_C_ 74.65 (C-4, Rha); moreover, H-4 of Rha (*δ*_H_ 4.95) was correlated with *δ*_C_ 172.29, suggesting that one acetyl was connected at C-4 of Rha. In the same way, another acetyl was substituted at C-2 of Rha, elaborated by the cross-peaks between *δ*_H_ 2.02 and *δ*_C_ 171.96/*δ*_C_ 73.46 (C-2, Rha) and between *δ*_H_ 5.29 (H-2, Rha) and *δ*_C_ 171.96. Compared to **2**, it was further confirmed by the up-field shifts of *δ*_H_ 5.29 (H-2 of Rha, ∆*δ*_C_ + 1.2 ppm) and *δ*_H_ 4.95 (H-4 of Rha, ∆*δ*_C_ + 1.43 ppm). The *β* configuration of D-apiose affirmed the chemical shifts of *δ*_C_ 112.54 (C-1), 78.49(C-2), 80.68(C-3), 75.41 (C-4), and 65.56 (C-5) [20]. The *a* configuration of L-rhamnopyranosyl was confirmed by the chemical shifts of Rha C-5 at *δ*_C_ 67.34 [21]. The anomeric proton coupling constants of L-arabopyranose (*J* = 7.6 Hz > 7.0 Hz) suggests that the configuration was an *α* orientation [19]. Thus, compound **3** was determined as 15-oxo-18-nor-(20*R*,22*R*,25*R*)-spirost-5,13-diene-1*β*,3*β*,21,23*α*,24*β*-pentol-1-*O*-*β*-d-apiofuranosyl-(1→3)-2,4-diacetyl-*α*-l-rhamnopyranosyl-(1→3)-*a*-l-arabinopyranoside.

Compound **4**, named Pamaiosides D, a white amorphous solid, was positive to Liebermann Burchard and Molisch chemical reactions. The pseudomolecular ion peak was measured in the HR-ESI-MS spectrum at *m/z* 993.3969 [M + Na]^+^ (calculated for C_46_H_66_O_22_Na, 993.3943), corresponding to the molecular formula C_46_H_66_O_22_. Compared with **3**, only two distinctions, the position of one acetyl group and the sugar linkages, were detected (Table 1, Table 2, Table 3 and Table 4). The acetyl group replaced at C-21, which was evidenced by the altered proton chemical shifts at *δ*_H_ 3.46 (H-4 of Rha, ∆*δ*_C_ -1.49 ppm), *δ*_Ha_ 4.19 (Ha-21, ∆*δ*_C_ + 0.45 ppm), and *δ*_Hb_ 4.33 (Hb-21, ∆*δ*_C_ + 0.54 ppm). It was further acknowledged by the cross-peaks between *δ*_H_ 2.08 (3H, *s*, C**H_3_**CO-)/*δ*_Ha_ 4.19, *δ*_Hb_ 4.33 (H-21), and *δ*_C_ 172.90 (CH_3_**C**O-) in the HMBC spectrum (Figure 5). In addition, the correlations for Api H-1 (*δ*_H_ 5.18) to Rha C-3 (*δ*_C_ 77.91), Rha H-1 (*δ*_H_ 5.28) to Ara C-4 (*δ*_C_ 75.62), and Ara H-1 (*δ*_H_ 4.30) to C-1 (*δ*_C_ 85.33) in the HMBC spectra clarified the linkages. Thus, compound **4** was characterized as 15-oxo-18-nor-(20*R*,22*R*,25*R*)-spirost-5,13-diene-21-*O*-acetyl-1*β*,3*β*,21,23*α*,24*β*-pentol-1-*O*-*β*-d-apiofuranosyl-(1→3)-2-acetyl-*α*-l-rhamnopyranosyl-(1→4)-*a*-l-arabinopyranoside.

Compound **5**, named Pamaiosides E, a white amorphous solid, was positive to Liebermann Burchard and Molisch chemical reactions. The pseudomolecular ion peak was measured in the HR-ESI-MS spectrum at *m/z* 935.3892 [M + Na]^+^ (calculated for C_44_H_64_O_20_Na, 935.3889), corresponding to the molecular formula C_44_H_64_O_20_. Compared to **3**, the proton signals at H-21 were replaced by one angular methyl, *δ*_H_ 1.16 (3H, *d*), in an aglycone moiety. Moreover, one keto-methyl at 21.27 (3H, *s*, *δ*_H_ 2.16) and one carbonyl at *δ*_C_ 173.78 signals were observed in the ^13^C-NMR spectra (Table 1, Table 2, Table 3 and Table 4). By analyzing the HMBC spectrum, the cross-peaks between *δ*_H_ 2.16 (C**H_3_**CO-)/*δ*_H_ 5.31 (H-24) and *δ*_C_ 173.78 (CH_3_**C**O) conjectured that one acetyl was substituted at C-24, and the up-field chemical shifts at H-24 (Δppm + 1.98) proved the hypothesis (Figure 6). According to the methodology, C-1, C-3, C-23, and C-24 possessed the same configuration as compound **3**, and the configurations of C-20, C-22, and C-25 were decided as *S*, *S*, and *R*, respectively. Therefore, the aglycone of **5** was determined as 15-oxo-18-nor-(20*S*,22*S*,25*R*)-spirost-5,13-diene-24-acetyl-1*β*,3*β*,23*α*,24*β*-tetrol.

Acid hydrolysis and GC analysis of **5** exhibited L-Ara, L-Rha, and D-Xyl residues in a ratio of 1:1:1. The configuration of each monosaccharide was deduced by the same approach employed in compound **1**, which was *α-*L-Ara, *α-*L-Rha, and *β-*D-Xyl, respectively. The sequence was derived from the correlations from Xyl H-1 (*δ*_H_ 4.44) to Ara C-4 (*δ*_C_ 85.51), Rha H-1 (*δ*_H_ 5.35) to Ara C-2 (*δ*_C_ 74.40), and Ara H-1 (*δ*_H_ 4.31) to C-1 (*δ*_C_ 85.57). Thus, compound **5** was identified as 15-oxo-18-nor-(20*S*,22*S*,25*R*)-spirost-5,13-diene-24-acetyl-1*β*,3*β*,23*α*,24*β*-tetrol-1-*O*-*β*-d-xylopyranosyl-(1→4)-[*α*-l-rhamnopyranosyl-(1→2)]-*a*-l-arabinopyranoside.

The five known steroidal saponins, **6**–**10**, were defined as 25(*R*)-spirost-5-en-1*β*,3*β*,21,23*α*,24*β*-pentol-1-*O*-*β*-d-apiofuranosyl-(1→3)-*α*-L-rhamnopyranosyl-(1→2)-[*β*-d-xylopyranosyl-(1→4)]-*a*-l-arabinopyranoside (**6**) [21]; 15-oxo-18-nor-25(*R*)-spirost-5,13-diene-1*β*,3*β*,21,23*α*,24*β*-pentol-1-*O*-*β*-d-apiofuranosyl-(1→3)-*α*-l-rhamnopyranosyl-(1→2)-[*β*-d-xylopyranosyl-(1→3)]-*a*-l-arabinopyranoside (**7**) [24]; 15-oxo-18-nor-25(*R*)-spirost-5,13-diene-24-acetyl-1*β*,3*β*,23*α*,24*β*-tetrol-1-*O*-*β*-d-apiofuranosyl-(1→3)-*α*-l-rhamnopyranosyl-(1→2)-[*β*-d-xylopyranosyl-(1→3)]-*a*-l-arabinopyranoside (**8**) [25]; 15-oxo-18-nor-25(*R*)-spirost-5,13-diene-1*β*,3*β*,21,23*α*,24*β*-pentol-1-*O*-*β*-d-apiofuranosyl-(1→3)-*α*-l-rhamnopyranosyl-(1→2)-[*β*-d-xylopyranosyl-(1→4)]-*a*-l-arabinopyranoside (**9**) [26]; 25(*R*)-spirost-5-en-1*β*,3*β*,21,23*α*,24*β*-pentol-1-*O*-*β*-d-*α*-l-rhamnopyranosyl-(1→2)-[*β*-d-xylopyranosyl-(1→3)]-*a*-l-arabinopyranoside (**10**) [27] (Table 5 and Table 6) by comparison of the physical and spectroscopic data available in the literature.

The discovery of the new compounds **1**–**5** extend the diversity and complexity of the spirostane saponin family. The cytotoxicity of **1**–**5** was evaluated against human pancreatic adenocarcinoma PANC-1 and BxPC3 cell lines using the CCK8 method. Regrettably, none of compounds showed significant cytotoxicity (Table 7).

## 3. Materials and Methods

### 3.1. General

Optical rotations were measured on a Perkin-Elmer 241 MC digital polarimeter (German PerkinElmer Corporation, Boelingen, Germany). 1D and 2D-NMR spectral experiments were measured in CD3OD on a Bruker AVANCE-500 and a Bruksmer AVANCE-800 spectrometer (Bruker Corporation, Karlsruhe, Germany) with TMS as an internal standard. The IR spectra were recorded on a Shimadzu IRPrestige-21 spectrophotometer (Shimadzu Corporation, Tokyo, Japan). The ESI-MS and HR-ESI-MS spectra were carried out on a Waters Micromass Quattro mass spectrometer (Waters, Shanghai, China). Column chromatographies (CC) were operated on a Sephadex LH-20 (GE-Healthcare, Uppsala, Sweden), ODS silica gel (Lichroprep RP-18, 40–63 µm, Merck Inc., Darmstadt, Germany), and silica gel H (10−40 µm, Qingdao Marine Chemical Inc., Qingdao, China). The GC analysis was performed on an Agilent 6890N apparatus using an HP-5 capillary column (30 m × 0.32 mm, 0.5 µm) and an FID detector with an initial temperature of 120 °C for 2 min and then temperature programming to 280 °C at the rate of 10 °C/min. Standards for D-xylopyranose (D-Xyl), L-arabopyranose (L-Ara), and L-rhamnose (L-Rha) were purchased from Sigma Chemical Co. (St. Louis, MO, USA), and D-apiose (D-Api) was purchased from Herbest Bio-Tech Co. (St. Baoguo, Baoji, China).

### 3.2. Plant Material

The rhizomes of *Paris mairei* were collected from Lijiang, Yunnan Province, China, in September 2018 and identified by the corresponding author Haifeng Tang. The voucher sample (No. 20180903) was deposited in the Department of Chinese Materia Medica and Natural Medicines, School of Pharmacy, Air Force Medical University, Xi’an, China.

### 3.3. Extraction and Isolation

The dried rhizomes of *Paris mairei* (1.0 kg) were chopped and refluxed with 70% ethanol (10.0 L) thrice (each 2 h). The ethanol solution was mixed and condensed with a vacuum rotary evaporator to receive a syrupy residue (584.0 g). The extraction was suspended in water (3.0 L) and extracted with same volume of petroleum ether and water saturated n-BuOH 3 times, successively. The water saturated in the n-BuOH layer was vacuum evaporated to give a gummy residue (132.0 g). The crude extraction was separated by silica gel column chromatography and eluted by gradient eluent of CH_2_Cl_2_-MeOH-H_2_O (100:0:0, 50:1:0, 20:1:0, 8:1:0.1, 6:1:0.1, 8:2:0.2, 7:2.5:0.1, and 6.5:3.5:0.1) to offer 13 fractions (Fr.1–13) based on the TLC analysis. Fr.13 was separated by silica gel column chromatography and eluted by a gradient eluent of CH_2_Cl_2_-MeOH-H_2_O (8:1:0.1, 8:2:0.2, 7:2.5:0.1, and 6:3:0.1) to get Fr.13-1 (1.1 g) and Fr.13-2 (830 mg). Fr.13-1 was eluted by MeOH on a Sephadex LH-20 to get rid of pigmentum and separated to Fr.13-1-1 (64 mg), Fr.13-1-2 (57 mg), and Fr-13-1-3 (145 mg) on ODS silica gel. Then, Fr.13-1-1 and Fr.13-1-3 were isolated by semi-preparative HPLC using MeCN-H_2_O (35:65, 40:60) as the mobile phase at a flow rate of 8.0 mL/min to afford compound **1** (9.1 mg, t*_R_* = 24.3 min) and **4** (8.8 mg, t*_R_* = 48.6 min), respectively. Fr.11 was eluted by MeOH on a Sephadex LH-20 to remove pigmentum to receive Fr.11-1 (4.2 g), Fr.11-2 (5.0 g), and Fr.11-3 (430 mg). Fr.11-2 was subjected to ODS silica gel and purified by a semi-preparative HPLC using MeCN-H_2_O (50:50) as the mobile phase at a flow rate of 8.0 mL/min to afford compound **3** (5.7 mg, t*_R_* = 44.1 min) and compound **7** (7.6 mg, t*_R_* = 40.2 min). Fr.12 was eluted by CH_2_Cl_2_-MeOH (20:80) on a Sephadex LH-20 to remove pigmentum and subjected to ODS silica gel to obtain Fr.12-1 (125 mg) and Fr.12-2 (670 mg). Then, compound **2** (26.7 mg, t*_R_* = 21.0 min) was offered by semi-preparative HPLC using MeCN-H_2_O (60:40) as the mobile phase at a flow rate of 8.0 mL/min. Fr.9 was purified by MeOH on a Sephadex LH-20 and separated on ODS silica gel to obtain Fr.9-1 (231 mg), Fr.9-2 (102 mg), and Fr.9-3 (193 mg). The three collections were successively purified by semi-preparative HPLC using MeCN-H_2_O (50:50, 40:60, 40:60) as the mobile phase at a flow rate of 8.0 mL/min to obtain compounds **5** (11.5 mg, t*_R_* = 35.3 min), **8** (5.5 mg, t*_R_* = 38.4 min), and **9** (4.6 mg, t*_R_* = 28.7 min). Fr.10 was eluted by CH_2_Cl_2_-MeOH (50:50) on a Sephadex LH-20 to remove pigmentum and subjected to ODS silica gel to obtain Fr.10-1 (75 mg) and Fr.10-2 (100 mg). Fr.10-1 and Fr.10-2 were isolated by semi-preparative HPLC using MeCN-H_2_O (75:25) as the mobile phase at a flow rate of 8.0 mL/min to afford compounds **10** (7.5 mg, t*_R_* = 18.3 min) and **6** (24.4 mg, t*_R_* = 21.6 min), respectively. The purity of all compounds was assessed by HPLC as being more than 95%.

### 3.4. Compound Characterization Data

Pamaiosides A (**1**): white amorphous solid, [α]_22D_ − 95.0 (c 0.05, MeOH); IR (KBr) ν_max_ (cm^−1^): 3420, 2930, 1080, 990, and 840; positive ESI-MS *m/z* 995.13 [M + Na]^+^, negative ESI-MS m/z 971.28 [M − H]^−^; positive HR-ESI-MS *m/z* 995.4824 [M + Na]^+^ (calculated for C_48_H_76_O_20_Na, 995.4828); ^1^H-NMR (800 MHz, CD_3_OD) and ^13^C-NMR (201 MHz, CD_3_OD) data, see Table 1.

Pamaiosides B (**2**): white amorphous solid, [α]_22D_ − 98.2 (c 0.06, MeOH); IR (KBr) *ν_max_* (cm^−1^): 3422, 2932, 1078, 988, and 840; positive ESI-MS *m/z* 1043.49 [M + Na]^+^; positive HR-ESI-MS *m/z* 1043.4677 [M + Na]^+^ (calculated for 1043.4675 C_48_H_76_O_23_Na); ^1^H-NMR (500 MHz, CD_3_OD) and ^13^C-NMR (125 MHz, CD_3_OD) data, see Table 2.

Pamaiosides C (**3**): White amorphous solid, [α]_22D_ − 105.2 (c 0.10, MeOH; IR (KBr) ν_max_ (cm^−1^): 3420, 2930, 1668, 1078, 989, and 842; positive ESI-MS *m/z* 993.48 [M + Na]^+^; negative ESI-MS *m/z* 969.28 [M − H]^−^; positive HR-ESI-MS *m/z* 993.3932 [M + Na]^+^ (calculated for 993.3943 C_46_H_66_O_22_Na); ^1^H-NMR (800 MHz, CD_3_OD) and ^13^C-NMR (201 MHz, CD_3_OD) data, see Table 3.

Pamaiosides D (**4**): White amorphous solid, [α]_22D_ − 110.0 (c 0.10, MeOH); IR (KBr) ν_max_ (cm^−1^): 3432, 2922, 1664, 1080, 990, 837; Positive ESI-MS *m/z* 993.18 [M + Na]^+^; Negative ESI-MS *m/z* 969.36 [M − H]^−^; Positive HR-ESI-MS *m/z* 993.3969 [M + Na]^+^ (calcd. for C_46_H_66_O_22_Na, 993.3943); ^1^H-NMR (800 MHz, CD_3_OD) and ^13^C-NMR (201 MHz, CD_3_OD) data, see Table 4.

Pamaiosides E (**5**): White amorphous solid, [α]_22D_ − 109.0 (c 0.15, MeOH); IR (KBr) ν_max_ (cm^−1^): 3430, 2925, 1080, 990, and 840; positive ESI-MS *m/z* 935.39 [M + Na]^+^; positive HR-ESI-MS *m/z* 935.3892 [M + Na]^+^ (calculated for C_44_H_64_O_20_Na, 935.3889); ^1^H-NMR (500 MHz, CD_3_OD) and ^13^C-NMR (125 MHz, CD3OD) data, see Table 5.

All the NMR (1D and 2D) and MS spectra of compounds **1**–**5** could be found in Appendix A (Appendix A).

### 3.5. Acid Hydrolysis and GC Analysis of the Sugar Moieties in Compounds ***1***–***5***

The assay was performed according to the procedure of Qiang F., et al. [28] with slight modifications, using N-(trimethylsilyl)imidazole as derivatization substrate. Compounds **1**–**5** (each 2 mg) were mixed with 2 mol/L CF_3_COOH (2 mL) and heated in a sealed tube at 110 °C for 8 h. Distilled water (20 mL) was added when the reaction was over and extracted with EtOAc (20 mL) three times. The aqueous layer was concentrated in vacuo by repeated mixing with methanol until the solvent was completely evaporated. The residue was dissolved in a 1 mL pyridine solution of 2 mg/L of L-cysteine methyl ester hydrochloride. After warming at 60 °C for 1 h, the solvent was evaporated under N_2_ protection. The reaction products were dissolved in the mixed solution of 0.2 mL N-(trimethylsilyl)imidazole and 2 mL anhydrous pyridine, and the mixture was warmed at 60 °C for another 1 h. Then, the solvent was evaporated under N_2_ protection. The residue was suspended in cyclohexane and water, the cyclohexane layer was the trimethylsilyl ether derivatives of monosaccharide. The mixture was filtered through a 0.45 µm membrane to remove the precipitate and analyzed by GC. Separations were carried out on an HP-5 capillary column (30 m × 0.32 mm, 0.5 µm). Highly pure N_2_ was used as a carrier gas (1.0 mL/min flow rate), and the FID detector operated at 250 °C (column temperature 250 °C). The carbohydrates were determined by comparing the retention times with standard trimethylsilyl ether derivatives prepared from authentic sugars using the same procedure performed for the sample. Retention times for authentic sugars after being derivatized were 11.23 min (D-Api), 12.20 min (L-Ara), 13.34 min (D-Xyl), and 14.48 min (L-Rha), respectively.

### 3.6. Cytotoxicity Assay for Compounds ***1***–***5***

The human pancreatic adenocarcinoma PANC-1 and BxPC3 cell lines were purchased from the Cell Bank of the Chinese Academy of Science (Shanghai, China) and cultured in DMEM (Corning, Beijing, China) supplemented with 10% FBS (Sigma, Shanghai, China) and 1% Penicillin–Streptomycin (Sigma, Shanghai, China) at 37 °C with 5% CO_2_. In the exponential phase of the growth, cells were plated onto 96-well plates at a concentration of 8000 cells/well for 24 h. Compounds **1**–**5** were prepared to various concentrations (80, 40, 20, 10, 5, 2.5, 1.25, and 0.625 µM in medium containing less than 0.1% DMSO) and incubated in 96-well plates (each concentration in six-fold wells) for 72 h. Gemcitabine (Gem, Meilunbio, ≥98%, Dalian, China) was offered as the positive control. Cell viability was determined according to reported assay methods using the commercial CCK8 kit (Elabscience, Wuhan, China) [29]. The optical density (OD) of each well was measured with an AMR-100 microplate reader at 450 nm (Allsheng Corporation, Hangzhou, China). Cytotoxicity emerged as the value of the drug concentration at the inhibition of cell growth by 50% (IC_50_).

## 4. Conclusions

This study afforded 10 compounds from the rhizomes of *Paris mairei*, including five new spirostane saponins. None of the new compounds exhibited cytotoxicity against PANC-1 and BxPC3 pancreatic cell lines, implying that the polyglucosides at 1-hydroxy in spirostane saponins may significantly decreased the activities of antitumor.

## Figures and Tables

**Figure 1 molecules-26-06366-f001:**
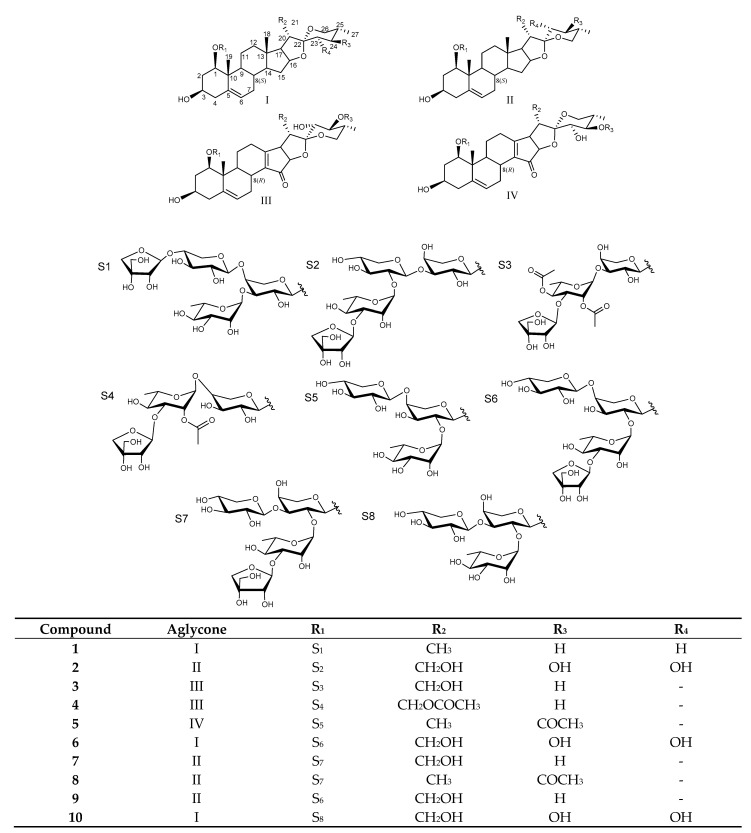
Structures of compounds **1**–**10**.

**Figure 2 molecules-26-06366-f002:**
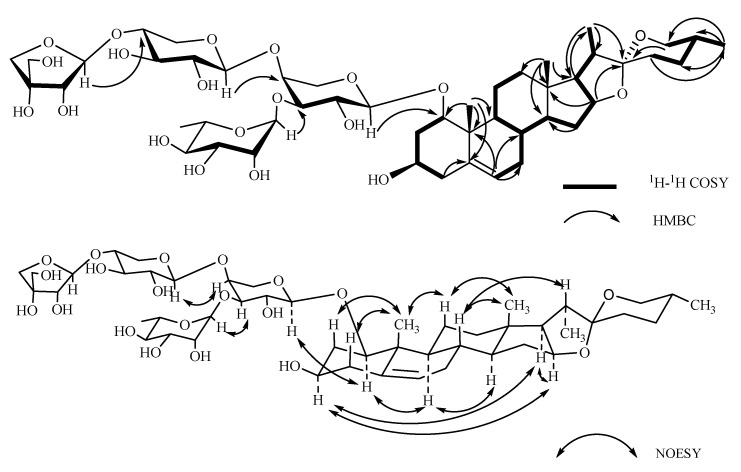
Key ^1^H-^1^H COSY, HMBC, and NOESY correlations of compound **1**.

**Figure 3 molecules-26-06366-f003:**
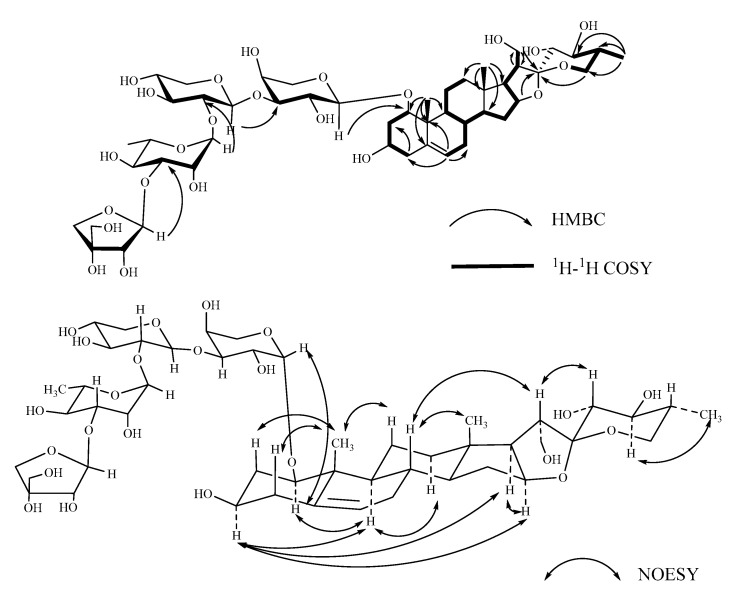
Key ^1^H-^1^H COSY, HMBC, and NOESY correlations of compound **2**.

**Figure 4 molecules-26-06366-f004:**
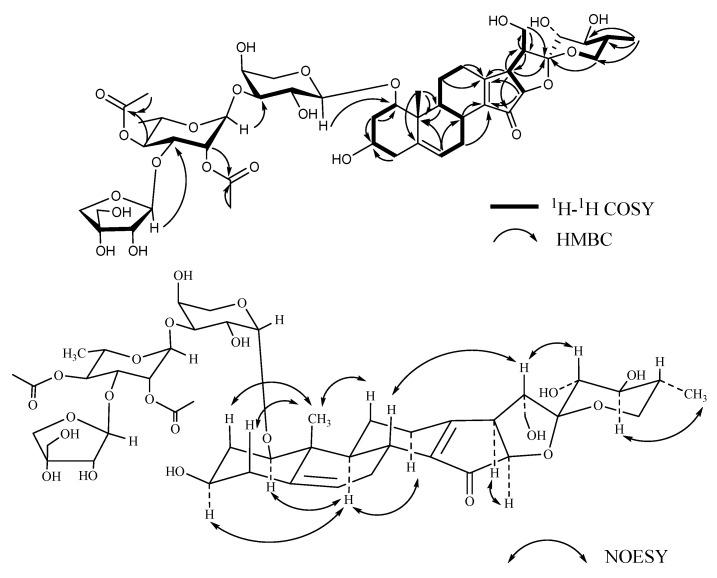
Key ^1^H-^1^H COSY, HMBC, and NOESY correlations of compound **3**.

**Figure 5 molecules-26-06366-f005:**
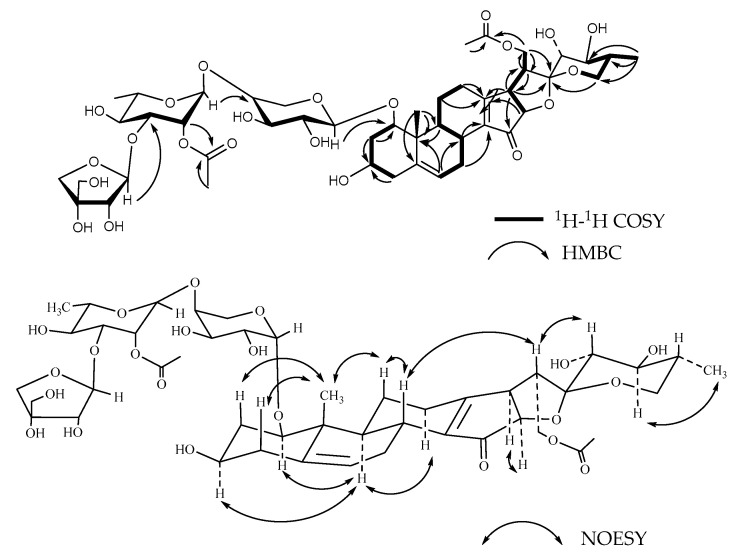
Key ^1^H-^1^H COSY, HMBC, and NOESY correlations of compound **4**.

**Figure 6 molecules-26-06366-f006:**
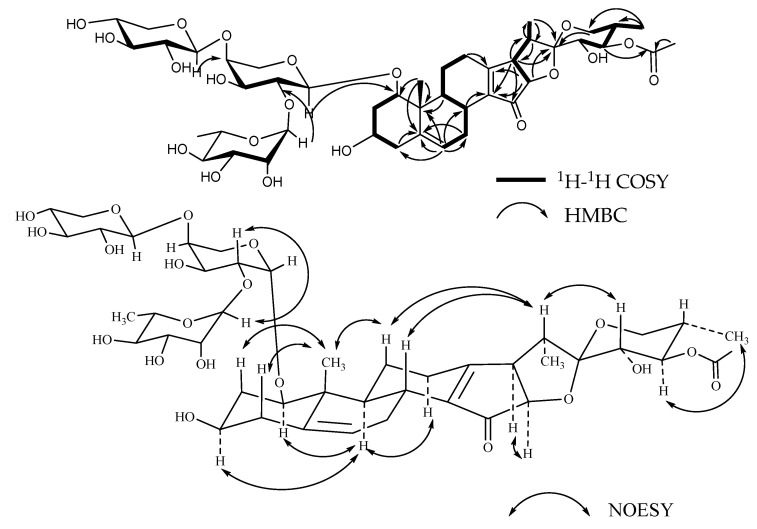
Key ^1^H-^1^H COSY, HMBC, and NOESY correlations of compound **5**.

**Table 1 molecules-26-06366-t001:** ^13^C-NMR data of aglycone moieties for compounds **1**–**5** in CD_3_OD.

Number	Compounds (*δ*_C_)				
	1 ^a^	2 ^a^	3 ^b^	4 ^b^	5 ^a^
1	84.79	84.80	85.28	85.33	85.57
2	37.45	37.46	37.88	37.66	37.55
3	69.37	69.33	69.19	69.37	69.34
4	43.52	43.52	42.80	42.91	43.04
5	139.71	139.79	139.82	139.69	139.70
6	126.17	126.12	126.42	126.19	126.15
7	33.04	32.86	30.32	30.27	30.27
8	34.26	34.27	32.88	32.93	32.79
9	51.54	51.53	48.91	48.86	48.76
10	43.58	43.61	43.36	43.30	43.31
11	24.95	24.84	26.24	26.27	21.27
12	41.40	41.24	29.17	29.09	29.28
13	41.29	41.80	179.23	178.36	179.36
14	58.11	58.26	139.51	140.38	139.58
15	32.85	33.13	207.09	206.45	207.11
16	82.38	84.65	82.38	82.19	82.92
17	64.14	58.63	49.66	49.84	52.38
18	17.65	17.30	-	-	-
19	15.52	15.53	14.35	14.31	14.38
20	43.05	46.04	49.54	46.32	43.66
21	15.07	62.95	61.99	64.53	14.08
22	110.74	112.72	114.60	114.25	113.48
23	32.59	71.20	74.51	74.48	67.53
24	30.04	74.01	76.27	76.10	73.87
25	31.59	36.57	39.33	39.47	35.39
26	67.97	61.39	65.81	65.90	62.38
27	17.29	12.97	13.29	13.25	12.52
				21-*O*-acetyl	24-*O*-acetyl
1	-	-	-	172.90	173.78
2	-	-	-	21.12	21.27

^a^ Tested in ^13^C-NMR (125 Hz); ^b^ tested in ^13^C-NMR (201 Hz).

**Table 2 molecules-26-06366-t002:** ^1^H-NMR data of aglycone moieties for compounds **1**–**5** in CD_3_OD.

Number	Compounds [*δ*_H_ mult.(*J* in Hz)]				
	1 ^a^	2 ^a^	3 ^b^	4 ^b^	5 ^a^
1	3.37 m	3.40 m	3.44 m	3.40 m	3.40 m
2	1.70 m, 2.11 m	1.72 m, 2.14 m	1.78 m, 2.17 m	1.78 m, 2.12 m	1.80 m, 2.11 m
3	3.34 m	3.38 m	3.43 m	3.38 m	3.39 m
4	1.90 m	2.22 m, 2.27 m	2.22 m, 2.27 m	2.24 m	2.25 m
5	-	-	-	-	-
6	5.56 br s	5.58 br s	5.63 br s	5.61 br s	5.62 br s
7	1.30 m	1.53 m, 1.97 m	1.48 m, 2.87 m	1.46 m, 2.87 m	1.47 m, 2.84 m
8	1.56 m	1.58 m	2.26 m	2.25 m	2.24 m
9	1.25 m	1.25 m	1.47 m	1.46 m	1.47 m
10	-	-	-	-	-
11	1.42 m, 2.54 m	1.44 m, 2.55 m	1.19 m, 2.94 m	1.19 m, 2.98 m	2.16 m
12	1.22 m, 1.65 m	1.19 m, 1.72 m	2.36 m, 2.60 m	2.38 m, 2.59 m	2.45 br s
13	-	-	-	-	-
14	1.15 m	1.78 m	-	-	-
15	1.91 m, 1.97 m	1.45 m, 2.02 m	-	-	-
16	4.38 m	4.53 q (7.50)	4.38 d (6.24)	4.40 d (6.24)	4.43 m
17	1.72 m	1.78 m	2.34 dd (6.64,14.48)	3.15 dd (6.56,7.84)	3.03 m
18	0.82 s	0.94 s	-	-	-
19	1.12 s	1.13 s	1.09 s	1.09 s	1.10 s
20	1.90 m	2.78 q (7.00)	3.14 m	2.50 m	2.08 m
21	0.96 d (6.90)	3.55 m, 3.69 m	3.74 m, 3.79 m	4.19 m, 4.33 m	1.16 d (6.9)
22	-	-	-	-	-
23	1.44 m, 1.73 m	3.52 m	3.87 m	3.33 m	3.56 m
24	1.62 m	3.76 m	3.33 m	3.34 m	5.31 t (2.9)
25	1.59 m	1.91 m	1.69 m	1.67 m	2.05 m
26	3.30 m, 3.43 m	3.32 m, 3.54 m	3.49 m, 3.52 m	3.50 m, 3.53 m	3.35 m, 3.73 m
27	0.80 d (6.35)	0.90 d (6.9)	0.93 d (6.56)	0.94 d (6.56)	0.79 d (6.90)
				21-*O*-acetyl	24-*O*-acetyl
1	-	-	-	-	-
2	-	-	-	2.08 s	2.16 s

^a^ Tested in ^1^H-NMR (500 Hz); ^b^ tested in ^1^H-NMR (800 Hz).

**Table 3 molecules-26-06366-t003:** ^13^C-NMR data of sugar portion of compound **1**–**5** in CD_3_OD.

Sugars	Compounds (*δ*_C_)				
	1 ^a^	2 ^a^	3 ^b^	4 ^b^	5 ^a^
Ara(*p*)					
1	101.16	101.13	101.32	101.37	101.62
2	74.58	74.56	71.22	71.11	74.40
3	80.45	85.25	74.60	75.98	70.67
4	85.29	70.52	76.16	75.62	85.51
5	67.04	67.03	67.93	67.75	67.21
Xyl					
1	106.47	106.44			106.44
2	74.91	74.89			74.88
3	78.04	78.01			78.07
4	70.54	70.78			71.19
5	67.04	67.03			67.02
Rha					
1	101.60	101.58	98.43	98.99	101.78
2	71.99	71.98	73.46	73.87	72.39
3	74.58	80.45	75.74	77.91	72.17
4	73.05	73.04	74.65	73.46	74.29
5	69.84	69.86	67.34	69.81	69.83
6	18.72	18.71	18.25	18.58	18.65
2-acetyl					
1	−	−	171.96	172.11	
2	−	−	21.02	21.09	
4-acetyl					
1	−	−	172.29	−	
2	−	−	21.15	−	
Api					
1	112.17	112.15	112.54	112.37	
2	78.23	78.28	78.49	78.34	
3	80.49	80.48	80.68	80.71	
4	75.18	75.17	75.41	75.35	
5	65.56	65.58	65.56	65.77	

^a^ Tested in ^13^C-NMR (125 Hz); ^b^ tested in ^13^C-NMR (201 Hz).

**Table 4 molecules-26-06366-t004:** ^1^H-NMR data of the sugar portion of compounds **1**–**5** in CD_3_OD.

Sugars	Compounds [*δ*_H_ mult.(*J* in Hz)]				
	1 ^a^	2 ^a^	3 ^b^	4 ^b^	5 ^a^
Ara(*p*)					
1	4.34 d (7.35)	4.34 d (7.35)	4.52 d (7.60)	4.30 d (7.44)	4.31 d (7.55)
2	3.83 m	3.83 m	3.74 m	3.73 m	3.86 m
3	3.70 m	3.76 m	3.76 m	3.69 m	4.00 m
4	3.76 m	3.99 m	3.72 m	3.73 m	3.76 m
5	3.22 m, 3.49 m	3.51 m, 3.86 m	3.52 m, 3.83 m	3.52 m, 3.83 m	3.54 m, 3.85 m
Xyl					
1	4.41 br d (7.10)	4.43 d (7.15)			4.44 d (7.16)
2	3.27 m	3.30 m			3.31 m
3	3.30 m	3.33 m			3.34 m
4	3.96 m	3.71 m			3.53 m
5	3.49 m, 3.84 m	3.51 m, 3.86 m			3.24 m, 3.88 m
Rha					
1	5.31 br s	5.33 br s	5.37 br s	5.28 m	5.35 br s
2	4.06 m	4.09 m	5.29 dd (1.76, 3.36)	5.28 m	3.93 br s
3	3.83 m	3.71 m	4.08 m	3.86 m	3.69 m
4	3.49 m	3.52 m	4.95 t (9.92)	3.46 m	3.41 m
5	4.11 m	4.14 dd (6.20, 9.55)	4.46 m	4.21 m	4.14 m
6	1.25 d (6.15)	1.27 d(6.15)	1.14 d (6.24)	1.26 d (6.16)	1.26 d (6.15)
2-acetyl					
1	−	−	−	−	−
2	−	−	2.02 s	2.08 s	−
4-acetyl					
1	−	−	−	−	−
2	−	−	2.12 s	−	−
Api					
1	5.19 d (2.90)	5.21 d (3.60)	5.04 d (2.24)	5.18 d (2.0)	
2	4.02 m	4.02 m	3.81 m	3.93 m	
3	−	−	−	−	
4	3.76 m, 4.07 m	3.79 m, 4.09 m	3.72 m, 3.93 m	3.72 m, 3.92 m	
5	3.62 m, 3.86 m	3.64 m	3.56 m, 3.65 m	3.55 m	

^a^ Tested in ^1^H-NMR (500 Hz); ^b^ tested in ^1^H-NMR (800 Hz).

**Table 5 molecules-26-06366-t005:** ^13^C-NMR data of the aglycone moieties of compounds **6**–**10**.

Number	Compounds				
	6 ^b^	7 ^a^	8 ^a^	9 ^a^	10 ^a^
1	84.74	85.37	85.42	85.37	84.98
2	38.18	37.56	37.62	37.56	37.45
3	68.87	69.35	69.40	69.38	69.28
4	44.38	42.94	42.96	42.95	43.59
5	140.08	139.73	139.72	139.72	139.80
6	125.40	126.16	126.16	126.22	126.10
7	32.58	30.26	30.29	30.31	32.86
8	33.75	32.82	32.82	32.88	34.25
9	50.94	48.77	48.74	48.80	51.56
10	43.51	43.33	43.34	43.36	43.59
11	24.60	26.26	26.38	26.25	24.80
12	40.99	29.28	29.29	29.21	41.24
13	41.46	179.29	179.38	179.31	41.79
14	57.61	139.89	139.60	139.85	58.26
15	33.05	207.37	208.15	207.08	33.14
16	84.11	83.15	82.96	82.36	84.63
17	58.37	48.77	52.40	49.37	58.61
18	17.58	—	−	—	17.32
19	15.69	14.41	14.40	14.41	15.50
20	46.25	50.13	43.69	49.49	46.03
21	62.99	62.03	14.06	61.98	62.94
22	113.19	114.18	113.50	114.84	112.71
23	70.99	70.64	67.56	74.51	71.19
24	73.65	73.32	73.89	76.25	73.96
25	36.61	36.38	35.42	39.30	36.55
26	61.31	61.97	62.40	65.80	61.39
27	13.63	12.97	12.52	13.29	12.98
24-*O*-acetyl					
1	−	−	173.78		
2	−	−	21.25		

^a^ Tested in CD_3_OD; ^b^ tested in C_5_D_5_N.

**Table 6 molecules-26-06366-t006:** ^13^C-NMR data of the sugar portion of compounds **6**–**10**.

Sugars	Compounds				
Ara(*p*)	6 ^b^	7 ^a^	8 ^a^	9 ^a^	10 ^a^
1	101.31	101.61	101.53	101.64	101.26
2	74.21	74.46	74.58	74.51	74.28
3	85.32	85.37	85.42	71.21	85.32
4	70.24	69.96	70.65	85.37	70.54
5	67.58	67.18	67.17	67.04	67.08
Rha					
1	102.02	101.47	101.69	101.47	101.71
2	72.39	71.96	72.00	71.98	72.38
3	80.40	80.46	80.48	80.46	72.12
4	73.17	73.01	73.06	73.03	74.42
5	70.13	69.80	69.84	69.81	69.87
6	19.63	18.73	18.72	18.73	18.64
Xyl					
1	107.16	106.50	106.54	106.52	106.49
2	75.23	74.88	74.92	74.90	74.90
3	78.95	78.05	78.09	78.05	77.99
4	71.59	71.19	71.23	70.65	70.73
5	67.58	67.02	67.04	68.18	67.02
Api					
1	112.31	112.13	112.18	112.16	101.26
2	78.36	78.24	78.27	78.25	74.28
3	80.83	80.48	80.48	80.48	85.32
4	75.73	75.16	75.18	75.18	70.54
5	66.17	65.56	65.59	65.59	67.08

^a^ Tested in CD_3_OD; ^b^ tested in C_5_D_5_N.

**Table 7 molecules-26-06366-t007:** Cytotoxic activity of compounds **1**–**5** against human pancreatic cancer cells in vitro (IC_50_, μM).

Compound	Cytotoxic Activity (IC_50_, μM; Mean ± SD, *n* = 3)	
	PANC-1	BxPC-3
**1**	>80	>80
**2**	>80	>80
**3**	>80	>80
**4**	>80	>80
**5**	>80	>80
Gemcitabine ^a^	0.0927 ± 0.0057	0.0376 ± 0.0031

^a^ Positive control.

## Data Availability

The data used to support the findings of this study are available from the corresponding author upon request.

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
