# Peer review of "New Steroidal Saponins Isolated from the Rhizomes of Paris mairei"

_molecules, 2021, doi:10.3390/molecules26216366_

Round 1

Reviewer 1 Report

In this manuscript, the authors isolated five new steroidal saponins together with five known compounds from the rhizomes of Paris mairei. The planner structures of new compounds were elucidated by NMR and MS techniques. The relative configurations were established by NOESY. In addition, the authors evaluated the cytotoxicity of new compounds against human pancreatic adenocarcinoma PANC-1 and BxPC3 cell lines.

The chemical structures of new compounds were not so novel, however, I recognize great efforts in this study.

The suggested structures should be correct, but it is advisable to confirm some of the structure determinations.

I think that this manuscript requires major revision before being ready for publication and the authors should correct the following:

  1. (Page 2, Figure 1) Compounds 3 and 4 having ester groups appear to be artifacts. Did the authors make sure they weren't artifacts?
  2. (Page 2, Figure 1) Since the carbon position number is not described, the structure determination part is difficult to understand. I recommend numbering the carbon position.
  3. (Page 2, Figure 1) The stereochemistry of aglycones I and II was not shown accurately. At least, the configurations of the C-8 position and the C-22 position should be shown.
  4. (Page 2, Figure 1, table) The positions of R3 and R4 are difficult to understand. In aglycone I, R3 is bound to C-23 position and R4 is bound to C-24 position, but in aglycone II, R4 is bound to C-23 position. I recommend that the authors correct them.
  5. (Page 2, Figure 1, table) I think R4 of compound 7 is wrong. Please check it.
  6. (Results and Discussion) I think that the structure of the aglycones of new compounds were not sufficiently determined. The analysis of HMBC correlation and NOESY correlation were fragmentary. If the aglycones can be isolated, the structure determination will be certain. I recommend isolating the aglycone moiety and performing structural analysis. If the authors cannot isolate the aglycones, please analyze HMBC correlation and NOESY correlation again. I think the following are necessary for the structural analysis of compound 1. I recommend that the structural analysis of the other new compounds also be performed in the same way as for compound 1.
  • (Page 4, Figure 2) In HMBC correlation of compound 1, the correlations for H-19 to C-9, H-18 to C-17, H-16 to C-22, H-26 to C-22, H-26 to C-22 were not shown. These correlations are important in determining the planar structure. Please check the correlations.
  • (Page 4, Figure 2) In NOESY correlation of compound 1, the correlations of H-3, H-8, H-16, H-17, and H-20 were not shown. These correlations are important in determining the relative configuration. Please check the correlations.
  • (Page 4, Figure 2) I recommend showing how to determine the configuration of C-22.
  1. (Page 4, Figure 2) I think that the structure of compound 1 in NOESY correlation is not correct. Please check and correct it.
  2. There are many mistakes such as a misspelling. I would point out some, but please review the whole once and make sure there are no mistakes.
  •  Please write everything below in italics.

       (Page 1, line 27) Paris

       (Page 1, line 30) Paris polyphylla, yannanensis, Paris polyphylla, Chinensis

  •  Please correct the following misspelling.

       (Page 1, line 38) Figrue 1. → Figure 1

  • (Table 1~5) Some parts are not well-organized and are not unified in style. Please check and correct them.
  •  (Page 11, line 205) Please delete “b-D-“.
  • (Page 15, 16) Please check the description method of the reference. There is no consistency.

Author Response

Dear Reviewer:

I have received your comments on our manuscript molecules-1411270 entitled “New steroidal saponins from the rhizomes of Paris mairei”. We appreciate for your examining and thanks a lot for your advisement.

According to your comments, I have made the recommended corrections and completed the revised manuscript in the attachment.  We look forward to hearing from you.

Reviewer 2 Report

The article submitted for review describes the structure of 5 new compounds belonging to a group of steroidal saponins. These compounds were isolated, along with 5 other known compounds, from rhizomes of Paris mairei and tested on selected cell strains.

The subject of the work and the way it is described make the article fit for publication in the “Molecules” and after minor corrections it could be published.

Remarks:

  1. No capital letters in the name of the University (line 11)
  2. The abstract is grammatically incorrect
  3. Words errors (e.g. Analysis line 16, figure line 38 and others)
  4. Frequent lack of capital letters or cursive in names of plants, compounds (e.g. Line 16, 27, 30, 236 and others)
  5. In figure 1, compound abbreviations should be bold
  6. A significant inconvenience is the lack of illustrations showing the numbering of atoms in the skeleton of the aglycone. This makes it very difficult to follow the description in the text.
  7. In NMR tables the columns should be separated between 1H NMR and 13C NMR data. This would make the results clearer and easier to read
  8. Lack of doi numbers in references; lack of complete data for part of the item
  9. Error in heading 3.6. (line 319), test results are presented only for compounds 1-5

Author Response

(The authors gave the same response as above.)

Round 2

Reviewer 1 Report

The manuscript has been revised well.

I think this manuscript will be acceptable after small corrections have been done.

  1. (Page 4, line 101) (Page 4, line 104) (Page 5, line 107) (Page 6, line 113) Please change [compound 1-5] to [compounds 1-5].
  2. (Page 7, line 128) (Page 8, line 157) (Page 9, line 190) (Page 10, line 208) Please change [Table 1-5] to [Tables 1-5].
  3. (Page 11, line 241) (Page 12, line 243) Please change [compound 6-10] to [compounds 6-10].